# National Experiences from 30 Years of Provider-Mediated Cascade Testing in Lynch Syndrome Families—The Danish Model

**DOI:** 10.3390/cancers16081577

**Published:** 2024-04-20

**Authors:** Lars Joachim Lindberg, Karin A. W. Wadt, Christina Therkildsen, Helle Vendel Petersen

**Affiliations:** 1The Danish HNPCC Register, Gastrounit, Copenhagen University Hospital—Amager and Hvidovre, DK2650 Hvidovre, Denmark; christina.therkildsen@regionh.dk; 2Department of Clinical Medicine, Faculty of Health and Medical Sciences, University of Copenhagen, DK2200 Copenhagen N, Denmark; karin.wadt@regionh.dk; 3Department of Clinical Genetics, Rigshospitalet, DK2100 Copenhagen Ø, Denmark; 4Medical Department, Zealand University Hospital, DK4800 Nykøbing Falster, Denmark; hellevp@regionsjaelland.dk; 5Clinical Research Centre, Copenhagen University Hospital, DK2650 Hvidovre, Denmark

**Keywords:** cascade genetic testing, direct contact, genetic counseling, Lynch syndrome, family communication

## Abstract

**Simple Summary:**

Cascade genetic testing is crucial for families with hereditary cancer syndromes like Lynch syndrome, where clinically established surveillance programs reduce illness and death. However, communicating genetic information within families faces barriers, with only a small uptake of genetic testing in family members informed through family-mediated contact. Provider-mediated interventions via letter or phone call increase testing uptake but raise legal and ethical concerns. We describe 30 years of national experience with cascade testing using unsolicited letters, covering administration, legislation, public attitudes, and testing rates. Our approach resulted in an average of 7.3 additional tests per family. Overall uptake of genetic testing was 54.4% after family-mediated and 64.9% after provider-mediated contact by letter. The uptake of genetic testing was highest in the first year after diagnosis of Lynch syndrome in the family, with 72.5% tested after provider-mediated contact. We propose a model for cascade genetic testing combining family- and provider-mediated contact.

**Abstract:**

Cascade genetic testing and surveillance reduce morbidity and mortality in Lynch syndrome. However, barriers to conveying information about genetic disorders within families result in low uptake of genetic testing. Provider-mediated interventions may increase uptake but raise legal and ethical concerns. We describe 30 years of national experience with cascade genetic testing combining family- and provider-mediated contact in Lynch syndrome families in the Danish Hereditary Non-Polyposis Colorectal Cancer (HNPCC) Register. We aimed to estimate the added value of information letters to family members in Lynch syndrome families (provider-mediated contact) compared to family members not receiving such letters and thus relying on family-mediated contact. National clinical practice for cascade genetic testing, encompassing infrastructure, legislation, acceptance, and management of the information letters, is also discussed. Cascade genetic testing resulted in 7.3 additional tests per family. Uptake of genetic testing was 54.4% after family-mediated and 64.9% after provider-mediated contact, corresponding to an odds ratio of 1.8 (*p* < 0.001). The uptake of genetic testing was highest in the first year after diagnosis of Lynch syndrome in the family, with 72.5% tested after provider-mediated contact. In conclusion, the Danish model combining family- and provider-mediated contact can increase the effect of cascade genetic testing.

## 1. Introduction

Lynch syndrome is an autosomal dominantly inherited cancer predisposition syndrome characterized by a significantly elevated risk of developing multiple primary cancers at an early onset. Colorectal and endometrial cancers are core Lynch syndrome cancers, but additional cancer types, such as upper urinary tract and small intestine cancers, are observed at an increased frequency. Internationally recognized surveillance programs, commencing at 25 years, have been proven to effectively reduce both the incidence of illness and mortality associated with Lynch syndrome [1,2,3,4]. It is crucial for individuals in Lynch syndrome families to gain knowledge about their potential cancer susceptibility, as identification of individuals carrying predisposing genetic variants plays a pivotal role in early diagnosis and cancer prevention through enrollment in surveillance programs [5].

As germline mismatch repair deficiency is usually inherited, it becomes essential to identify additional family members carrying the same pathogenic variant whenever an individual is diagnosed with Lynch syndrome. Hence, clinical guidelines advocate for cascade testing within families affected by Lynch syndrome [6,7,8,9]. In such a cascade-testing approach, the initial focus is to test first-degree relatives to the index person, defined as the first person to be diagnosed with Lynch syndrome in the family, as they are most likely to share the same pathogenic variant. Subsequently, the process “cascades” through the family as new carriers of the pathogenic variant are found, and genetic testing is offered to their first-degree relatives [6]. Cascade testing has demonstrated its effectiveness in identifying individuals who carry the pathogenic variant but have not yet developed cancer [10,11]. However, the success of cascade testing hinges on how effectively the risk information is conveyed to other family members at risk. This conveying can occur either through communication from a family member who has been diagnosed with the pathogenic variant (family-mediated contact) or through healthcare professionals (provider-mediated contact) [11].

In many countries, the responsibility for conveying the risk information is primarily delegated to the index person due to legal restrictions that prevent healthcare professionals from directly contacting relatives [12,13,14]. This practice is very subjective and unreliable as it is influenced by a combination of individual, interpersonal, and environmental factors. Notably, concerns about familial closeness and worries that relatives may struggle to comprehend the implications of genetic test results have been cited as reasons for hesitating to share such information [5,14]. Consequently, family-mediated cascade testing is reported to result in around 36% of the relevant family members being genetically tested, leaving many family members unaware of their potential risks and opportunities for prevention [11]. In contrast, it has been demonstrated that healthcare providers can increase the proportion of genetically counseled and tested family members [11,15]. However, the provider-mediated approach requires legal rights, supportive infrastructure, ethical considerations, and acceptance among professionals and family members [16,17].

To the best of our knowledge, Denmark stands out as the only country in the world with three decades of experience in implementing direct provider-mediated contact within Lynch syndrome families. In this paper, we describe 30 years of national clinical practice for cascade genetic testing, covering aspects related to infrastructure, legal frameworks, and the acceptance of this approach. Additionally, we assess the added value of provider-mediated contact on the uptake of genetic testing (the Danish experience). Finally, we propose a model for cascade genetic testing combining family- and provider-mediated contact (the “Danish model”).

## 2. Materials and Methods

### 2.1. The Danish Healthcare System

Denmark, with a population of nearly 6 million inhabitants, has a healthcare system that encompasses genetic counseling, genetic testing, surveillance procedures, and cancer treatment, all funded through taxes and provided free of charge to citizens fulfilling clinical criteria, with very little variation across urban and rural regions. The waiting period for the index person of a family suspected of having a hereditary cancer syndrome to receive genetic counseling varies from 2 to 12 months. However, waiting times for subsequent family members are considerably shorter in most regions.

A cornerstone of the Danish healthcare system is the Central Population Register (CPR). Since 1 April 1968, every citizen in Denmark has been assigned a unique CPR number, and their names, date of birth, current addresses, and parent–child relationships are recorded in the register [18,19]. All information noted in the healthcare and public systems is linked to personal CPR numbers, including diagnoses, education, and income. Denmark has a long history of building and maintaining national registers, which can be linked through the CPR for research purposes. Some of these registers are supported by the government with mandatory reporting, while others, like the Danish Hereditary Non-Polyposis Colorectal Cancer (HNPCC) Register, are center-based and rely on voluntary data collection.

### 2.2. The Danish Legislation

The Danish legislation has changed over the 30 years the HNPCC Register has existed. Hence, the management of cascade testing conducted by the HNPCC Register has adjusted accordingly to follow current regulations. Initially, oral consent was obtained by the HNPCC Register from 1991 to 2016, but it transitioned to written informed consent (see Appendix A) to comply with the European General Data Protection Rules (GDPR) and Danish legislation introduced in 2017.

In Denmark, cascade genetic testing in hereditary cancer syndromes relies on specific regulations in the national legislation (https://dsmg.dk/wp-content/uploads/2022/11/Uopfordret-kontakt-til-risikopersoner.pdf (accessed 5 March 2024)) [20]. Inhere, medical doctors are allowed to directly contact a Danish citizen under the following conditions:(1)The person is at risk of a life-threatening condition;(2)There is an accurate test to clarify if the person has the condition;(3)The consequences of the condition can be avoided or reduced if diagnosed early;(4)The potential benefit for the person outweighs the potential harm inflicted by giving the information.

In case a family member declines contact from the HNPCC Register, no provider-mediated contact will be initiated.

### 2.3. The Danish HNPCC Register

The national HNPCC Register was established in 1991 at the Gastroenterology department (called the Gastro Unit, covering both gastrointestinal surgery and medical gastroenterology) at Copenhagen University Hospital, Hvidovre, by local gastroenterological surgeons with the aim to reduce morbidity and mortality from hereditary colorectal cancer [21]. Since then, specialized staff members have registered families with HNPCC (covering Lynch syndrome but also familial colorectal cancer with a yet unknown genetic cause) in a dynamic database, adding new data to family members, e.g., genetic counseling, genetic test results, and surveillance outcomes [21,22,23].

Until 2009, approximately 90% of the families included in the register were diagnosed locally at the HNPCC Register. Over the years, clinicians engaged locally at the HNPCC Register have actively promoted the voluntary reporting of data by colleagues from other hospitals. Today, the register receives data on approximately 20 newly diagnosed Lynch syndrome families per year from all six departments of clinical genetics in Denmark. The HNPCC Register is an integrated part of the Gastro Unit. It is managed by a senior scientist who bears responsibility for both the staff and the scientific research conducted. Operational expenses include salaries for one chief physician working halftime with the register, two part-time genetic assistants recording the reported data in the database, and a part-time data manager extracting data for research purposes and quality assessments. The registered data are hosted by a private company, which offers a secure electronic platform and data storage (https://hd-support.dk/product/winhnpcc/ (accessed 5 March 2024)) [24]. An annual fee covers support services and ongoing platform development. The platform was developed and implemented with external funding from an EU grant between 2003 and 2007 [25].

### 2.4. The Clinical Approach

Like the Danish legislation, the Danish healthcare system has changed during the last three decades, and the clinical approach for provider-mediated contact and cascade genetic testing has changed accordingly. In the first two decades, the HNPCC Register staff included geneticists and gastroenterological surgeons, while as of today, the register is more a source of knowledge and a coordinating center for clinical management at the local genetics departments, mediating data sharing across geographical regions and medical reporting systems. Now, the HNPCC Register and clinical genetics departments jointly expand the Lynch-syndrome family pedigrees and link parents and siblings to search for at-risk relatives via the CPR. At-risk relatives not linked in the CPR can be identified by searching their names and/or addresses (even previous addresses) in the CPR. At-risk relatives deceased before 1968 are searched for in census counts and church records, e.g., newborns and marriages, when relevant to identify living cousins and their offspring.

In the first report to the HNPCC Register, the genetic counselor identified family members at risk. The HNPCC Register sends out unsolicited information letters to individuals who are at risk at the first report. This procedure is not automated but is based on an assessment from an HNPCC staff member. As time passes, more genetic data may be reported on the family members, and the HNPCC Register monitors these data and strives to send out information letters to the new first-degree members at risk identified via the cascade testing. Thus, the register enables equitable access to genetic counseling for families residing in various regions across the country.

Until 2015, the HNPCC Register exclusively handled the dispatching of unsolicited letters to at-risk family members. With the establishment of a national oncogenic society, local genetic counselors occasionally distributed these letters. Each regional genetics department has established its own local guidelines for conveying information, with each department determining the allocation of time and resources for drafting and handing out or sending out information letters intended for patients to share with their family members. Letters dispatched from the clinical genetics departments are not systematically documented in the HNPCC Register. However, in most cases, genetics departments do not send unsolicited letters but rely on the HNPCC Register to send such information letters to specific relatives in agreement with the departments.

First-degree relatives younger than the age from which surveillance is recommended can be registered on a reminder list. This triggers a provider-mediated letter if a genetic test is not registered in the HNPCC Register three months after their recommended surveillance should have commenced. Second-degree at-risk relatives within the recommended surveillance age are contacted if the closest relative(s) are dead or genetic testing has not been performed. Third-degree relatives are only contacted if first- and second-degree relatives are deceased either at a young age (<60 years) or have been affected with a known Lynch syndrome-associated cancer.

In addition to the initial direct contact, the HNPCC Register can send out one reminder letter to at-risk relatives if the HNPCC Register has not received a copy of their genetic test result and at least two years have passed since they were last contacted. This reminder procedure is not performed systematically due to limited time and resources in the HNPCC Register. Instead, when new data, e.g., diagnosis or test results on the family, are registered, it is a part of the daily routine to check if data on the family are missing.

### 2.5. Definition of Study Cohort and Statistics

The first Lynch syndrome case in Denmark was genetically tested on 12 December 1993, defining the beginning of the study period, and the first information letter was sent from the national Danish HNPCC Register on March 16, 1995, defining the onset of provider-mediated contact. The end of the study was 26 April 2023, when all Lynch syndrome relatives at risk registered in the Danish national HNPCC Register with a valid CPR number were identified (*n* = 12,017, Figure 1). “Family members at risk” was defined as blood-related family members being either (1) carriers of a Lynch syndrome-defining pathogenic variant, (2) first-degree relatives to such carriers, (3) genetically untested relatives affected by a Lynch syndrome-associated cancer, or (4) genetically untested first-degree relatives to such affected patients. A few second-degree relatives were included if their relevant parent died younger than 60 years. Obligate non-carriers were not included.

We excluded the index person, defined as the first verified Lynch syndrome case, in each family (*n* = 638). Using data from the CPR, we removed relatives who were already deceased when the first case in the family was diagnosed (*n* = 2759), but we did not exclude family members who died during the 30 years of follow-up. We also excluded relatives living abroad when the first case in the family was diagnosed (*n* = 187). To secure data completeness, we excluded family members with less than two years of observation time after either receiving a letter (provider-mediated contact, *n* = 59) or after their family was diagnosed with Lynch syndrome (family-mediated contact, *n* = 387) or because they were younger than 27 years at the end of follow-up (*n* = 1480). This corresponded to an average of 18.8 identified, and 10.2 included family members at risk per family, covering relatives in the extended family identified by multiple cascades of genetic testing, reaching beyond the first- and second-degree relatives to the first carrier in the family.

To investigate the initial transmission of information and its subsequent passage to the next generation, we categorized the remaining 6507 eligible family members at risk into two groups: (1) individuals who were old enough to commence surveillance when Lynch syndrome was first diagnosed in the family (25 years or more), and (2) individuals who were either unborn or below the age of 25 years at the first diagnosis.

From the HNPCC Register, we extracted data on sex and dates of birth, death, the first diagnosis of Lynch syndrome in the family, dispatching of information letters, and genetic tests. We also collected data on results of genetic tests and country of residency.

In this paper, “provider-mediated contact” refers to the transmission of an information letter from the HNPCC Register to a Lynch syndrome family member. “Family-mediated contact” was defined here as all other methods of conveying information within the family, including intermediated contact by an information letter handed over from family members to their relatives.

The uptake of genetic testing was calculated as numbers and percentages of tested family members at risk, not including obligate carriers. The distribution of sex and age in each group was calculated as numbers and percentages. For family members receiving provider-mediated contact, “age at contact” was defined as their age when receiving the information letter, and “time to contact” was defined as the time to dispatching of the letter since the first diagnosis of Lynch syndrome in the family. For individuals not receiving a provider-mediated letter (family-mediated contact), data on the dissemination of information in the family were not available, so “age at first diagnosis” was defined as age at the time when the first member of the family was diagnosed with Lynch syndrome and “time to test” was defined as time to genetic testing since the first diagnosis of Lynch syndrome in the family. As the letters were sent in a dynamic manner, with some family members responding faster than others, we have no data on when the first round of cascade genetic testing ended and the next round began.

The effect of provider-mediated contact on cascade genetic testing was estimated as odds ratios (ORs) using multivariate logistic regression modeling comparing the uptake of genetic testing after provider- and family-mediated contact. The analyses were adjusted for sex and age, which were considered confounding factors for the uptake of genetic testing.

Statistical analyses were performed in SAS [26], and *p* values < 0.05 were considered significant. Multivariate logistic regressions were computed using the logistic procedure.

## 3. The Danish Experience from 30 Years of Cascade Genetic Testing

In total, 5491 family members at risk were 25 years or older at the time of the first Lynch syndrome diagnosis in the family, thus fulfilling the criteria for genetic testing (Figure 1). Hereof, 1873 (34.1%) had been informed via unsolicited letters from the HNPCC Register (provider-mediated contact) throughout the study period (Appendix A), and 3618 (65.9%) were not informed directly by letter but were alleged to have received oral or written information about their risk from family members (family-mediated contact). An additional 1016 family members at risk reached the age of 25 during the study period and were subsequently included. Hereof, 224 (22.0%) were informed by a letter from the HNPCC Register.

The combined family- and provider-mediated contact in the total cohort resulted in 7.3 subsequent genetic tests per family on average. Considering the cohort >25 years, 1967 (35.8%) of the individuals informed via the family were genetically tested, while the combined effect of information via letters and family reached an uptake of genetic testing of 57.9%. For the family members turning 25 years old during the study period, 564 (55.5%) were tested following family-mediated contact, while the combined effect of family- and provider-mediated contact resulted in 709 (69.8%) genetically tested family members. As mentioned above, letters were not sent to everybody, and to evaluate the effect and confounding factors of the letters, we divided the cohort into two subsets: one receiving an unsolicited letter and one relying solely on information from family members.

### 3.1. Uptake of Genetic Testing in Individuals ≥25 Years at First Lynch Syndrome Diagnosis in the Family

After family-mediated contact, 1967 (54.4%) family members at risk were genetically tested, and after provider-mediated contact, 1215 (64.9%) family members at risk were genetically tested, corresponding to an OR of 1.8 (95% CI 1.6–2.1, *p* < 0.001) when adjusting for sex and age.

We observed a significant influence on the uptake of genetic testing from sex, age, and time—and the associations were similar in both the family-mediated and the provider-mediated contact groups (Figure 2A–F). Notably, provider-mediated contact within the first year resulted in 72.5% (95% CI 69.6–75.5%) of the relevant family members being genetically tested, corresponding to an OR of 2.6 (95% CI 2.2–3.0)

### 3.2. Uptake of Genetic Testing among Individuals below 25 Years at First Diagnosis of Lynch Syndrome in the Family

At the genetic counseling, carriers of a pathogenic germline variant were encouraged to secure genetic testing of their children before the age of 25 years to ensure surveillance commencing at the recommended age according to guidelines. We found that 379 (37.3%) were genetically tested before the age of 25 years. Among the remaining 637 family members, 185 (47.7%) were genetically tested after family-mediated contact and 145 (58.2%) after provider-mediated contact, corresponding to an OR of 1.5 (95% CI 1.1–2.1, *p* = 0.009).

### 3.3. Untested Family Members

Guidelines for sending unsolicited letters have been implemented and revised during the study period, so some relatives at risk did not receive a letter when their family was diagnosed with Lynch syndrome. Due to limited resources and time to systematically revisit all the families after the continuous updates of the guidelines, not all 6507 at-risk family members have been tested. In the study period, 2531 relatives were tested after family-mediated contact, a letter was sent to 2097 relatives, and 344 had died. This left 1535 (23.6%) of the family members at risk, untested, and without provider-mediated contact.

### 3.4. The Danish Model for Cascade Genetic Testing

Based on clinical experiences and the results reported above, we recommend the following model for conveying risk information in families with Lynch syndrome (Figure 3):(1)Identify first-degree relatives at risk during genetic counseling with the index person;(2)Recommend conveying information to the relatives about their risk of Lynch syndrome, cancer risk, the possibility of risk-reducing surveillance, and how to be referred to genetic counseling (family-mediated contact);(3)Agree with the index person on a specific timeframe (e.g., two months) to convey the information;(4)Send a follow-up letter directly to the at-risk relatives ≥25 years and register relatives <25 years on a reminder list for a future letter (provider-mediated contact);(5)Repeat steps 1–4 whenever a new family member is identified with the pathogenic germline variant.

This model caters for relatives preferring to be informed by a close relative and secures correct and timely transmission of facts, which is a known concern with patient-mediated contact [27].

## 4. Discussion

This paper describes the Danish experiences from national cascade genetic testing in families with Lynch syndrome, as well as the necessary measures regarding the healthcare system, legal foundation, and clinical collaborations to implement such practice. We have summarized 30 years of real-world clinical efforts to contact at-risk family members in Lynch syndrome families, showing that the subset receiving unsolicited letters had a higher uptake of genetic testing with the odds of being genetically tested after provider-mediated contact at 1.8 compared to family-mediated contact.

To summarize our experiences, uptake of genetic testing was generally higher among women, which contrasts with previous studies [28], linearly associated with younger age, and higher within the first two years from the first Lynch syndrome diagnosis in the family. Following this, the effect of family-mediated contact decreased fast, while provider-mediated contact still yielded a high uptake of genetic testing. Furthermore, provider-mediated contact improved the uptake of genetic testing among children turning 25 years after the first diagnosis.

Cascade genetic testing is a continual and time-consuming process that should persist across generations. A simulated model from the United States has calculated the difference in time when using the family-mediated contact versus the provider-based approach [29]. Given a family size of three generations with 2–4 children per generation and using a panel of the 18 most common genes associated with hereditary cancer syndromes (including the three mismatch repair genes *MLH1*, *MSH2*, and *MSH6*), it would take 9.9 years to detect all carriers using the provider-mediated approach if 70% of the first-, second-, and third-degree relatives were tested. This number increases to 59.5 years if the cascade testing is performed via family-mediated communication and non-systematic testing [29], supporting the use of provider-mediated contact.

The Danish approach to cascade genetic testing resulted in an average of 7.3 additional genetic tests per family during the study period. This is somewhat higher than previously reported by the Finnish HNPCC Register, where 1.77 additional genetic tests were performed per index person [30,31]. In the Finnish model, genetic testing was solely performed when the relatives approached a genetic counselor based on information from a family member (usually a parent) due to a cancer history in a close relative or personal cancer diagnosis [27]. A systematic review from 2013, including eight studies on uptake from cascade genetic testing among Lynch syndrome families, reports low outputs (between 0.2 and 3.6), emphasizing (in line with other studies) barriers in knowledge dissemination and family communication [32,33,34]. A Dutch model from the Department of Clinical Genetics of the Erasmus MC University Medical Center has encouraged newly diagnosed Lynch syndrome individuals to share oral and written information with their relatives [35]. However, this merely family-mediated approach resulted in 3.6 additional tests per family. The low efficacy of cascade testing is likely to be explained by a lower range of individuals (30–52%) referred to genetic counseling, as the uptake of genetic testing is up to 95% once referred to genetic counseling [28,30].

Here, we suggest a model (the “Danish model”) where cascade genetic testing is facilitated through a strategy combining family- and provider-mediated contact to increase the efficiency as previously described [11,36] (Figure 3). To our knowledge, this specific approach has not been practiced in other countries or local registers, likely due to ethical and legal concerns. In Denmark, the only national registers for hereditary cancer syndromes are the HNPCC Register and the Familial Adenomatous Polyposis (FAP) Register. The two registers were initiated and developed simultaneously at the same gastroenterological department and were, for the first 20 years, managed by the same dedicated physicians. The lack of similar registers for other hereditary syndromes may be explained by the absence of political interest, devoted clinicians, and hospital resources. In addition, legal and ethical considerations regarding genetic information are continuously getting more complex as data sharing and personal genetic fingerprints reach media attention. Over the years, legislation on personal information has changed significantly towards a more restricted frame reflected by the European GDPR, which has also affected the clinical practice at the HNPCC Register, and Danish geneticists are now more reluctant to send unsolicited letters many years after the first Lynch syndrome diagnosis in the family.

To this end, ethical concerns have been raised regarding the acceptability of provider-mediated contact [16,37]. Though physicians may fear approaching family members unsolicited, evidence supports that cascade testing should move towards direct contact in families with hereditary cancer syndromes, and it is important to notice that this approach is supported by a growing number of studies on patient and public attitudes showing high rates of acceptability related to direct contact [12,27,38]. We have previously investigated the attitudes to provider-mediated contact in the form of unsolicited information letters in the Danish population and among Danish Lynch syndrome individuals receiving such letters [27]. In both groups, we found a high acceptance of provider-mediated contact, with only 3% preferring not to be informed about the risk of having Lynch syndrome. Around 40% found it very important to be informed by a close relative, and the majority preferred to receive information through a letter from a healthcare provider rather than from a distant relative.

The sharing of genetic test results hinges on various individual factors, encompassing a sense of responsibility to share information, the nature of the relationship, emotional closeness, personal emotions, and perceptions of how the information will be received [36,39]. While interventions like phone calls to at-risk relatives, educational websites, and online platforms have been tested to enhance cascade testing in relatives of probands, their effectiveness varies [32]. A growing argument supports a shift in the care delivery paradigm towards direct contact with relatives [12]. Individuals commonly feel they have a “right to know” about their potential risks and opportunities for preventing serious diseases [11,12,27], and direct contact from healthcare professionals can secure equal and objective distribution of risk information.

Based on experiences from Denmark, we contend that specific conditions should be considered before healthcare professionals directly contact potential at-risk family members. Firstly, professionals must possess the legal authority for such contact, and there should be a supportive infrastructure in place to identify at-risk relatives and establish comprehensive family pedigrees. Secondly, ethical considerations, including the delicate balance between an individual’s right to know or not to know and the benefits of surveillance versus the potential harm from awareness, must be thoroughly deliberated. Achieving a consensus on navigating these complex ethical issues is crucial [16,17], given that ethical concerns are strongly influenced by culture, social values, and religion. Therefore, when determining the approach to convey information, patient preferences must be taken into account, considering cultural and national variations in decision-making.

This study is a result of a real-world clinical setting where provider-mediated contact has been implemented during the study period. The study’s strengths include 30 years of solid data with almost no loss to follow-up due to the Danish CPR and the exclusion of the index person, minimizing the risk of overestimation. However, as the data were not systematically collected for research purposes, we may have introduced inherent biases, which may affect the generalizability of our findings.

As this study is based on data solely collected from the Danish HNPCC Register, limitations include missing reports of genetic test results to the register, missing reports of family members that declined genetic testing at genetic counseling, lack of knowledge about provider-mediated contact from the local genetics departments, and lack of knowledge regarding how and when the family-mediated contact was provided in the families. Information letters sent from the local genetics departments may result in an underestimation of the effect of the provider-mediated contact in this study. However, we do not consider such practice to impact the results significantly, as the clinical genetics departments, due to limited resources, heavily rely on the HNPCC Register to disseminate the information and very rarely send out these letters themselves. Furthermore, we know from previous studies that 44% of the family members receiving the information letter from the HNPCC Register already know of their risk from family-mediated contact [27]. However, as this study measures the overall impact of a combined model of information dissemination, we do not consider this a bias.

Though we suggest a cascade genetic testing strategy combining family- and provider-mediated contact, we acknowledge that contacting all relatives and revisiting the families continuously is a time-consuming procedure. In clinical practice, limited time and resources may be barriers to such a strategy, as we have also experienced in the Danish setting. However, to accommodate such barriers in Denmark and improve our outcome, we would need to carefully inspect each family pedigree to identify the remaining 1535 untested at-risk members and judge if they fulfill the current guidelines for sending unsolicited letters. Though we are currently trying to solve this problem, the inclusion of these data in the present study will result in an underestimation of our results.

## 5. Conclusions

Provider-mediated contact by letter to family members at risk adds value to the family-mediated contact and increases the uptake of genetic testing. We suggest that direct contact with family members should be included in clinical genetic caretaking of families with Lynch syndrome when possible.

## Figures and Tables

**Figure 1 cancers-16-01577-f001:**
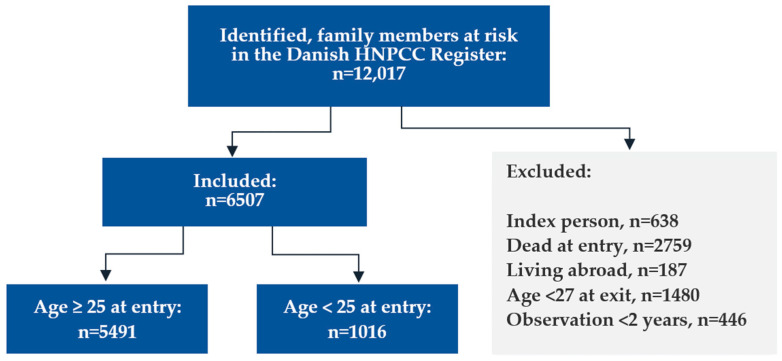
Flowchart showing in- and exclusions.

**Figure 2 cancers-16-01577-f002:**
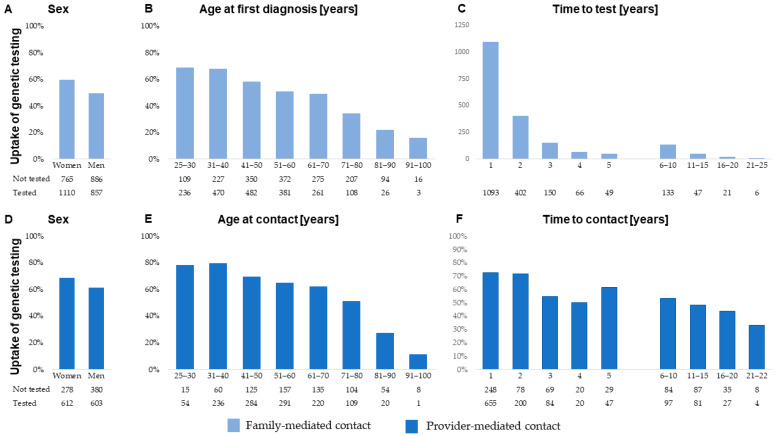
Uptake of genetic testing among at-risk family members aged 25 years or older at the time of the first Lynch syndrome diagnosis in the family stratified by family-mediated contact (upper panel) and (**A**) sex, (**B**) age at first diagnosis of Lynch syndrome in the family, and (**C**) time to test, and by provider-mediated contact (lower panel) and (**D**) sex, (**E**) age at provider-mediated contact, and (**F**) time to contact.

**Figure 3 cancers-16-01577-f003:**
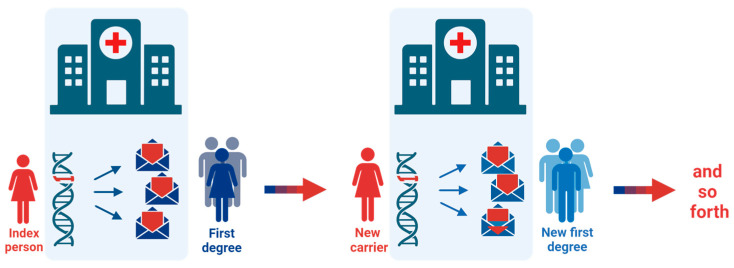
The Danish model for cascade genetic testing. When an index person is diagnosed with Lynch syndrome, all at-risk first-degree relatives are recommended and invited to genetic counselling by the index person and by mail from the healthcare provider. If new carriers of a pathogenic variant are identified in the family, invitations are sent to their relevant first-degree relatives. If first- or second-degree relatives are deceased either at a young age or have been affected with a Lynch syndrome-related cancer, invitation letters are sent to their first-degree relatives. The cascade testing continues as far as possible, primarily including first- and second-degree family members.

## Data Availability

The data presented in this study are available on request from the corresponding author. The data are not publicly available due to privacy restrictions.

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
