# Peer review of "National Experiences from 30 Years of Provider-Mediated Cascade Testing in Lynch Syndrome Families—The Danish Model"

_cancers, 2024, doi:10.3390/cancers16081577_

Round 1

Reviewer 1 Report

Comments and Suggestions for Authors

See attach PDF-document.

Reviewer 2 Report

Comments and Suggestions for Authors

Thank you for the opportunity to review this important paper. In order to be ready for publication, I recommend that the authors address the following issues:

1. There are English language errors throughout. I did not line edit these, but I have highlighted them in my copy of the manuscript if that would be helpful. They are mostly errors related to the usage of verb tenses, eg. saying "barriers to convey" when the correct expression is "barriers to conveying" and also errors in conjucation (eg. "Short interventions with provider-mediated contact have increased the uptake of genetic testing but raises concerns..." which should be "raise" concerns, because "interventions" is plural).

2. P2, lines 72-73, "index person" is not common parlance, either define it or use a more intuitive label such as "first family member to be diagnosed" or something like that. 

3. P2, lines 94-95 should be gastroenterological surgeons not "gastro" surgeons.

4. P3, line 100 the term "carrier" typically refers to an unaffected person, if this person had Lynch syndrome then you should say "the first Lynch syndrome case" or something like that.

5. P4, lines 147-164, this section describing the context of the Danish healthcare system should be provided in the background section, not the results section. 

6. P5, lines 171-174: The source for the Danish legislation should be cited. 

7. P5, line 192: It is interesting that you suggest uniform genetic counseling is desirable when in fact personalized medicine involves tailoring of genetic counseling to individual clinical information. In Lynch syndrome specifically, risk differs enormously depending on the gene in which there is a pathogenic variant. It does not seem appropriate to assume that uniform genetic counseling is a desired goal. 

8. P4, lines 195-200: It is unclear whether the letter being sent is "triggered" by an automated mechanism, or if there is staff that manually sends these letters. This is important because it has implications for the generalizability of the approach to settings with fewer human resources. If the letter is automated or if it is manually sent, please specify this. 

9. P5 lines 209-216: "Acceptability" and ethical or legal concerns are not the same thing. The paper does not clearly elaborate what the ethical concerns are regarding the provider-mediated approach. This is a missing section of the paper. If the data presented in this section is about acceptability, then do not say it is also about law and ethics. The fact that most people who received an unsolicited letter did not object to it is useful to know, as it suggests that members of the Danish public do not share the ethical concerns that have been raised, but the Danish context is also one where everyone has healthcare access and there seems to be relatively high trust in public services, so it is not clear that this conclusively addresses the deeper ethical concerns that have been raised about the provider mediated approach. Also, more detail is required to better understand the study design and the difference between the two groups whose attitudes were explored. 

I will be happy to see this paper again once the above issues have been addressed. 

Comments on the Quality of English Language

The English language errors throughout the manuscript are all fairly common mistakes that occur when Danish is translated into English. Most of them are issues with verb tenses or conjugation and should not be difficult to fix. 

Round 2

Reviewer 1 Report

Comments and Suggestions for Authors

Please see enclosed file
